# Low temperature self-densification of high strength bulk hexagonal boron nitride

Haotian Yang[1], Hailiang Fang[2], Hui Yu[1], Yongjun Chen[1], Lianjun Wang [2], Wan Jiang[2], Yiquan Wu[3] & Jianlin Li[1]

Hexagonal boron nitride (hBN) ceramics are expected to have wide applications at high temperatures as both a structural and functional material. However, because of its flake structure and general inertness, it is currently impossible to sinter hBN powder to a dense bulk (with a relative density of above 96%) even at 2000 °C. Here, we report dense bulk hBN with 97.6% theoretical density achieved at a lower preparation temperature (1700 °C) via a self-densifying mechanism without sintering additives. During the sintering process, cubic boron nitride particles incorporated into the hBN flake powders transform into BN onions with a significant volume increase, thus filling in voids among the hBN flakes and highly densifying the hBN bulks. The resulting dense hBN ceramics possess 2–3 times the strength of traditional hBN ceramics. This phase-transition-induced volume expansion strategy could lead to dense sintered compacts with high performance in other ceramic systems.

[1] State Key Laboratory of Marine Resource Utilization in South China Sea & School of Materials and Chemical Engineering, Hainan University, No. 58 Renmin Ave, Haikou 570228, China. [2] State Key Laboratory for Modification of Chemical Fibers and Polymer Materials & College of Materials Science and Engineering, Donghua University, Shanghai 201620, China. [3] Kazuo Inamori School of Engineering, New York State College of Ceramics, Alfred University, 2 Pine Street, Alfred, NY 14802-1296, USA. Correspondence and requests for materials should be addressed to L.W. (email: wanglj@dhu.edu.cn) or to Y.W. (email: wuy@alfred.edu) or to J.L. (email: jlli@hainu.edu.cn)

Hexagonal boron nitride (hBN) ceramics are widely-used materials with excellent properties, characterized by its excellent refractory behaviour at high-temperatures, chemical inertness, high lubricity, excellent thermal shock resistance, and favourable dielectric properties[1–4]. hBN ceramics exhibit high thermal conductivity (130–150 W m$^{-1}$ K$^{-1}$ parallel to the plane and 22–25 W m$^{-1}$ K$^{-1}$ in the perpendicular direction)[5], a low dielectric constant (4.0–5.4) and a low dielectric loss tangent ($3 \times 10^{-4}$)[1]. However, the generally low density and weak mechanical strengths attainable by current consolidation techniques reduce the applicability of the material. As such, the main challenge with hBN ceramics is to improve the density of hBN ceramics, which seems a challenging task to overcome using conventional consolidation practices.

hBN has a typical layered structure similar to graphite, with an equal number of alternating B and N atoms that are covalently bonded by sp$^2$ hybridization within a two-dimensional layer. However, the material also exhibits remarkable ionic character[6], resulting in electrons in the π bond that cannot be delocalized. As such, hBN is not a conductor like graphite, where the 2D layers are held together by weak van der Waals forces[7], which also allows the layers to easily slide over each other. The special layered structure of hBN makes it difficult for the hBN flakes to stack densely and the strong in-layer covalent bonding results in the sluggish self-diffusion during sintering. All these hinder the densification of the hBN ceramics. Conventional pressureless sintering of hBN powder without any additives results in ceramics of less than 70% relative density[8–10]. Therefore, hot-pressing (HP) is typically adopted to prepare dense hBN ceramics, although sintering additives are still required to achieve suitable densities.

A variety of sintering additives have been explored to promote the densification of boron nitride ceramics, including B$_2$O$_3$ and a number of other oxides (Table 1). Although the addition of B$_2$O$_3$ does promote densification by providing a low melting liquid that enhances diffusion during sintering, the residual B$_2$O$_3$ is detrimental to the material's resistance to water attack and its high temperature properties. Other oxide additives are used that react with the B$_2$O$_3$ on the hBN flake surface to form an oxide glass liquid that facilitates consolidation. When a significant amount of these additives are used, they also act as reinforcements to enhance the materials' mechanical properties, as summarized in Table 1.

Although oxide additives are beneficial in achieving higher densities and better mechanical properties in hBN ceramics, their addition significantly harms the dielectric and thermal properties of the material. hBN ceramics produced with sintering aids cannot be used at extremely high temperatures, and their resistance to chemical attack is greatly reduced. As such, the addition of sintering additives eliminates the favourable properties that make boron nitride a unique and attractive material in the first place.

Recently, Zhang et al.[11] sintered hBN without any additives via spark plasma sintering (SPS) under an applied pressure of 100 MPa and achieved a relative density of 95.1% at a temperature of 2000 °C. While Zhang's results are encouraging, the sintering conditions are too harsh to be feasible for the mass production of dense, high-purity dense hBN ceramics. Additionally, while the achieved density is an improvement on past results, significant room for improvement still exists.

All previous studies seemingly indicated that the upper limit of the relative density of hBN ceramics is approximately 95% and that the preparation of dense, additive-free hBN ceramics is all but an insoluble problem, still, efforts have been made. In our recent research[12], we sintered diamond powders at 1600 °C by SPS and obtained graphite onion bulks with superior strength. These graphite onions were derived from nano-diamond particles. This achievement has made us draw upon an alternative route to synthesize high strength hBN (HSHBN) bulk materials using BN onions produced in situ from cubic boron nitride (cBN) as sintering aids and reinforcements.

cBN, a polymorph of hBN, has very similar crystal structure and lattice constant as diamond and transforms to hBN at approximately 1400 °C under ambient pressure conditions (more information in Supplementary Note 1)[13–15]. When heated to a high temperature above 1400 °C, cBN particles gradually transform into a structure of the BN concentric-shell onion morphology[16].

Herein, we report the preparation of pure, dense and high strength hBN ceramics through the sintering of cBN/hBN powder mixtures via SPS (Fig. 1, Table 2). As cBN has a significantly higher density (3.48 g cm$^{-3}$) than hBN (2.27 g cm$^{-3}$), the transformation of cBN particles to BN onions during sintering is accompanied by a significant volumetric expansion. The expanding material serves to both fill the voids between larger hBN particles, and subsequently reduce diffusion distances and allows it to assist in the densification process. With this method, we achieve low-temperature sintering of high density and high strength BN bulk materials without any additives. The strategy is efficient and opens up new possibilities for applications in many other material systems.

## Results

**Microstructure.** The X-ray diffraction (XRD) patterns of the raw materials used in this study and the sintered HSHBN bulks with 0–100 wt% cBN sintered at 1700 °C for 5 min are shown in Fig. 2a. Six diffraction peaks detected are apparent in the XRD pattern of HSHBN samples at 26.63°, 41.51°, 43.75°, 50.00°, 54.85° and 75.73° 2$\theta$, correspond to the (002), (100), (101), (102), (004) and (110) reflections of hBN, respectively[17]. No diffraction peaks corresponding to cBN, located at 43°, 50° and 74° 2$\theta$, can be detected, indicating that all of the starting cBN has transformed to hBN during sintering.

During conventional uniaxial hot pressing of common hBN, the hBN flakes become preferentially oriented because of the flake structure of the hBN particles, which produces pronounced anisotropy in end ceramics[5]. This preferred orientation can be substantiated by XRD analysis. The anisotropy of hBN ceramics is apparent in its strong (002) reflection (Fig. 2a), which declines

---

**Table 1 hBN ceramics with different additives**

| Additive(s) | Processing | Relative density | Side effect |
|---|---|---|---|
| 4 wt% B$_2$O$_3$[2] | HP | 92% | Water-resistance and high-temperature refractoriness reduced |
| 4.5 wt% B$_2$O$_3$[32] | SPS, 1600 °C, 25 MPa | 94.7% | |
| 5 wt% Al$_2$O$_3$/CaO[33] | HP, 1800 °C, 40 MPa | 94.1% | High-temperature refractoriness reduces and dielectric loss increased |
| 20 wt% mullite[7,34] | HP, 1900 °C, 30 MPa | 94% | |
| 22 wt% Y$_2$Si$_2$O$_7$[35] | HP, 1740 °C, 25 MPa | 85% | |
| None[11] | SPS, 2000 °C, 100 MPa | 95.1% | None |

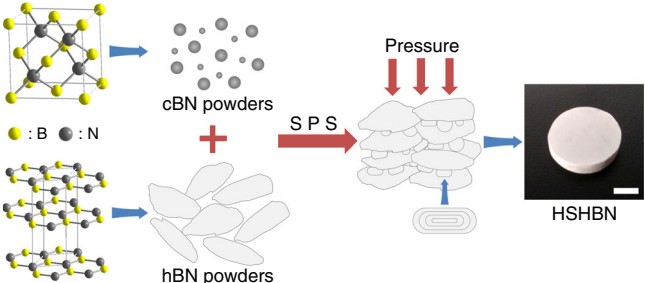

**Fig. 1** Schematic illustration of the HSHBN consolidation process. The scale bar represents 5 mm

**Table 2 A list of the examined compositions**

| Sample code | wt% cBN powders (200 nm) | wt% cBN powders (400 nm) | wt% hBN powders (1–2 μm) |
|---|---|---|---|
| hBN ceramics | 0 | 0 | 100.00 |
| HSHBN-15 | 3.75 | 11.25 | 85.00 |
| HSHBN-30 | 7.50 | 22.50 | 70.00 |
| HSHBN-50 | 12.50 | 37.50 | 50.00 |
| HSHBN-80 | 20.00 | 60.00 | 20.00 |
| HSHBN-100 | 30.00 | 70.00 | 0 |

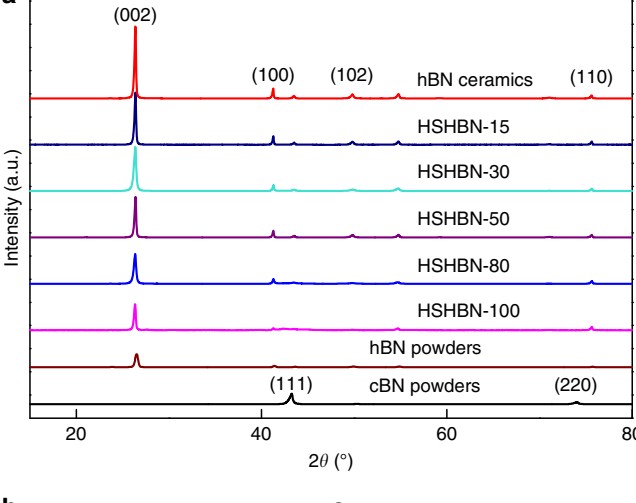

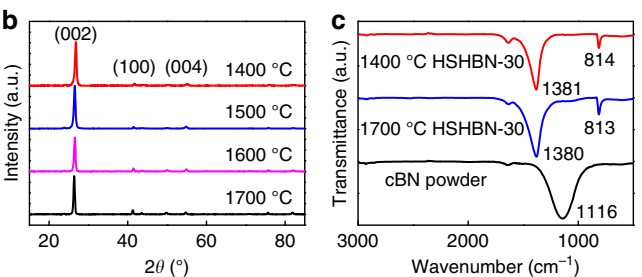

**Fig. 2** Characterization of phase evolution in the as-prepared samples. XRD patterns of the cBN and hBN precursor powders, pure hBN ceramics and HSHBN samples sintered at 1700 °C (**a**); XRD patterns of HSHBN-30 with sintering temperatures of 1400 °C–1700 °C (**b**); Fourier transform infrared spectroscopy (FTIR) spectra of cBN powder, and HSHBN-30 samples sintered at 1400 °C and 1700 °C (**c**)

continuously with increasing cBN addition, as the onions formed from cBN during sintering have an isotropic microstructure. The HSHBN-30 samples sintered at temperatures of 1400 °C–1700 °C were also analysed by X-ray diffraction (Fig. 2b). The XRD patterns of all the HSHBN-30 samples sintered at different temperatures show that the cBN sintered at or above 1400 °C has been completely transformed to BN onions.

Figure 2c shows the room temperature FTIR spectra of cBN powder, 1400 and 1700 °C sintered HSHBN-30. The spectrum of HSHBN-30 contains two strong peaks at approximately 1379 and 813 cm$^{-1}$, which are assigned to B–N stretching vibrations and B–N–B bending vibrations, respectively, as detailed by previous reports[18,19]. The weak peak at 1632 cm$^{-1}$ is the stretching vibration peak of O–H in $H_2O$. The 1116 cm$^{-1}$ peak attributed to cBN cannot be detected[20], an observation that supports the results of the XRD analysis (Fig. 2a) indicating that the cBN phase completely disappears during sintering. Additionally, the peak attributable to O–B–O vibrations at 490 cm$^{-1}$ could not be detected (Fig. 2c), suggesting that near-zero amount of $B_2O_3$ is present in the HSHBNs[17].

Transmission electron microscopy (TEM) micrographs of different regions of HSHBN-30 are shown in Fig. 3. Two different morphologies are presented in Fig. 3a. One features the layered structure of hBN flakes, while the other shows the BN onions squashed by the applied pressure during SPS. The high resolution transmission electron microscopy (HRTEM) images clearly show the parallel lattice fringes of the hBN flakes and the curved lattice fringes of the concentric BN onions. The TEM observations also provide some insight into aspects of the cBN to hBN phase transformation of during sintering. The BN onions appear deformed, likely by the applied pressure of the sintering process, with the outer layers of the onions being squeezed and entwined together to form strong bonds between the BN onions and hBN flakes.

The inset of Fig. 3c shows the selected area electron diffraction (SAED) pattern collected from the sample of HSHBN-30. The fuzzy and discontinuous ring diffraction patterns manifest the poly-crystalline nature of the HSHBN. The rings correspond to the (002) and (100) planes of hBN, respectively, in good agreement with the XRD results. These results demonstrate that cBN has completely transformed to BN onions.

**Sintering mechanism.** The sintering of powders is usually associated with volumetric shrinkage of the compact due to the elimination of voids between particles. To achieve a high packing density and reduce pores in the powder compact, particle size gradation is very important. Smaller particles are added to fill the voids between larger particles to obtain a high packing density. Horsfield filling theory[21] addresses the relationship between the particle size distribution, volume fraction and porosity of mixed powder compacts. Based on this rule, two different cBN powder particle sizes (200 nm and 400 nm) were added a ratio of 1:3 to fill the voids between hBN flakes to achieve a high packing density.

The relative densities of the HSHBN samples are shown in Fig. 4. The measured-relative densities of pure hBN and pure cBN sintered at 1700 °C are approximately 93 and 88%, respectively. Relative density was observed to increase with increasing cBN content, reaching a maximum at 30 wt% cBN. The relative density of HSHBN-30 reached 95.5% at 1600 °C, slightly higher than the highest-reported result for hBN ceramics, of 95.1% in a sample sintered at 2000 °C[11]. When the sintering temperature is increased to 1700 °C, the relative density of the HSHBN-30 correspondingly increases to 97.6%. Supplementary Figure 2 shows the field emission scanning electron microscopy (FESEM) micrographs on polished surfaces of the samples.

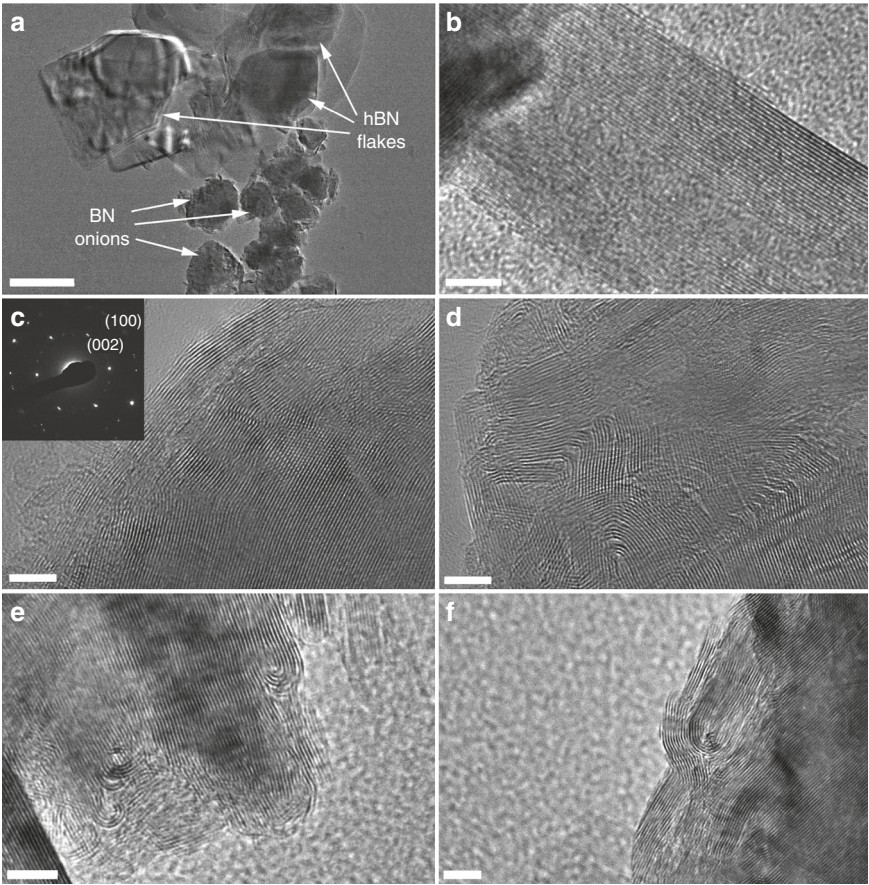

**Fig. 3** TEM images of HSHBN-30 sample. TEM image of squashed BN onions and hBN flakes (**a**); HRTEM images of hBN flakes (**b**); BN onion associated SAED pattern (**c**), deformed and entwined BN onions (**d**), and the curly hBN layer shells stripped from the BN onions (**e**, **f**). The scale bar is 500 nm in **a**, and represents 5 nm in **b**–**f**

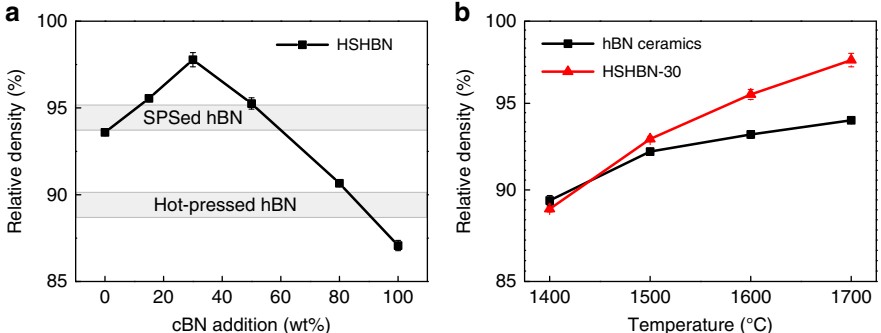

**Fig. 4** Relative densities of HSHBNs and pure hBN ceramic. Relative densities of the bulk HSHBNs sintered at 1700 °C with different cBN contents, with the highest relative densities reported from other studies shown in the shadowed areas (**a**), and the temperature dependence on the relative density of pure hBN ceramics and HSHBN-30 (**b**). All error bars represent standard deviations, and $n = 5$

For traditional powder consolidation, the sintering process is composed of three stages[22]. The initial stage consists of particle rearrangement, followed by volumetric shrinkage. Then, particle neck growth occurs, and densification proceeds via mass transport and atom diffusion. During the third stage, pore elimination and rapid grain growth occur. During SPS, the grain growth typically taking place during the third stage is reduced by the short time interval over which the process is conducted; however, coarsening is not completely inhibited.

In the traditional phase transformation-assisted sintering of TiO$_2$[23], two major driving forces contribute to the sintering. One is the enthalpy change of the anatase to rutile transformation, and the enthalpy of anatase-rutile transformation is 6.51 kJ mol$^{-1}$. The other is the release of surface energy due to grain growth. However, in our present work, no significant grain growth is observed that reduces the surface energy. The enthalpy change ($\Delta_{tr} H_{c\text{-}h}$) of the cBN-hBN phase transition was calculated according to the Kirchhoff's law. In the case of little change in pressure, $\Delta_{tr} H_{c\text{-}h}$ (1673.15 K) = 15.4867 kJ mol$^{-1}$, i.e., 15.4867 kJ mol$^{-1}$ of heat is absorbed at 1400 °C (Supplementary Note 2). It is manifested that cBN itself has no release of energy as a sintering driving force in the phase transformation process at 1400 °C. Therefore, other densification mechanisms must exist.

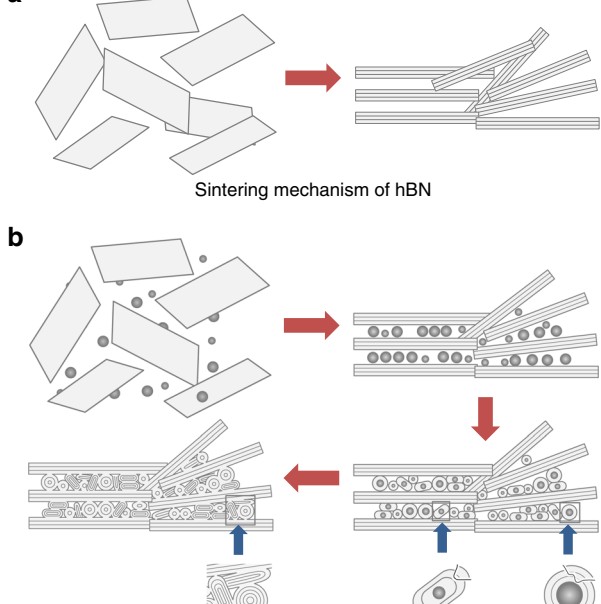

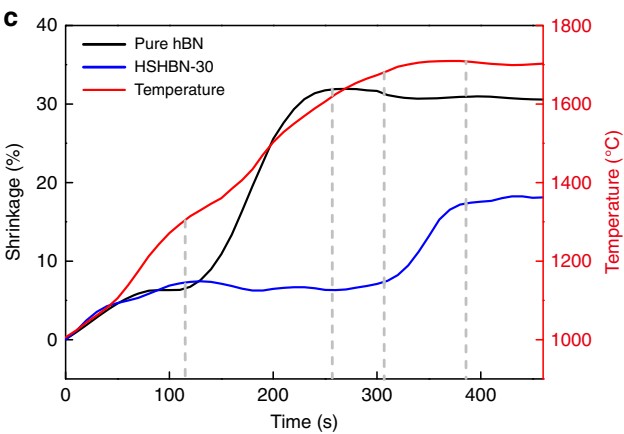

**Fig. 5** Consolidation mechanisms of hBN and HSHBN during SPS. Sintering mechanisms of pure hBN (**a**) and HSHBN (**b**), and sintering shrinkage curves of pure hBN and HSHBN-30 above 1000 °C (**c**)

have been deformed and curled under pressure are shown in Fig. 3e, f. Understandably, BN onions are to be squashed by the intense stresses to further compact densely.

With the transformation proceeding, a volume expansion associated with the phase transformation continues, and more empty space becomes occupied by the expanding BN onions. The outer layers of the BN onions also become entwined with the layers of the hBN flakes as they grow, forming a semi-interlocking structure.

Last but not least, during the phase transformation, the B and N atoms forming the cBN particles are very active, and the resultant rapid atomic diffusion is beneficial to the densification and elimination of pores among particles[23,24]. More importantly, the highly active B and N atoms create strong bonding between hBN flakes and BN onions to form a robust bulk.

Figure 5c shows the shrinkage curves of pure hBN and HSHBN-30 samples. Because of the cBN-derived BN onions that hinder the sintering of hBN powder, the shrinkage of HSHBN-30 starts at ~1650 °C while densification happens to the pure hBN sample at ~1350 °C. The distinct feature is the significantly smaller overall shrinkage of the denser HSHBN-30 (~16%) compared to that of the less denser, pure hBN (~30%), strongly demonstrating that the volumetric expansion associated with the transformation of cBN to hBN makes a major contribution to the densification of HSHBN-30.

As shown in Fig. 4a, the relative densities of the two pure samples are much lower than those of the doped ones, further substantiating the notion that the addition of cBN to form BN onions assists in the filling of the pore space between hBN flakes. On the one hand, the cBN particles enhance the sintering by producing a larger volume of hBN. On the other hand, these tiny cBN particles act as a second phase that hinders the sintering of hBN flakes. However, too many BN onions will form a rigid connecting network that hinders the consolidation. Thus, a balance has to be reached to produce the densest material, which is HSHBN-30 in this case.

In summary, during consolidation, the transformation of cBN particles to BN onions plays a key role by forming an additional volume of hBN that acts as a sort of glue bonding the hBN flakes together, which is a spontaneous and self-densifying process.

Obviously, a temperature increase enhances the sintering of hBN flakes, which occurs both in pure and doped hBN compacts. As shown in Fig. 4b, a higher temperature enhances the sintering of HSHBN-30 more significantly than that of pure hBN ceramics. In fact, a higher temperature facilitates the deformation of BN onions that contributes to the densification by eliminates avoids among the spherical BN onions. For example, according to the established Coble or Nabarro-Herring diffusion mechanisms, B and N atoms migrate along or across a BN onion from the onion top under the compressive stress to the onion lateral that is under tensile stress, which strongly depends on the temperature in addition to local stresses. In contrast, when a compact of only hBN powder is sintered, atoms have to diffuse over relatively long distances to reach regions where they will fill voids.

The sintering mechanism of HSHBN is presented in Fig. 5b. The phenomena occurring in all three stages of sintering must be considered to understand the mechanism by which the HSHBN sintering process occurs. First, the hBN flakes and cBN particles are rearranged to form a denser compact, due to the effects of applied pressure. To achieve this, the smaller cBN particles must partially fill the voids between hBN flakes, thus increasing the packing density. As the temperature is ramped up to 1400 °C, the cBN particles gradually transform to BN onions. It is at this point that the self-densifying mechanism of the cBN begins to occur. cBN has a much greater density than hBN, and thus upon transformation of cBN to hBN, a phase transition-induced volume expansion of approximately 50% occurs. Due to the geometry of the onions, the greatest stresses will occur at the outside of the particle, and thus the transformation will occur from the outside of the cBN particle and gradually moves inwards. During SPS, as the powders are confined in the die and due to the intensive local stresses occurring at the contact points between cBN particles, the hBN shell layers formed early in the process are essentially extruded into the pores between flakes, assisting in the densification process. Such hBN layer shells that

**Mechanical behaviour.** To understand the mechanical behaviours of the produced HSHBN materials, compression, flexural and nanoindentation tests were performed on polished samples. For the boron nitride samples sintered at 1400 °C, because the lubricity of hBN makes it compact more easily. However, flexural tests revealed the higher density does not necessarily reflect a more advanced degree of sintering in the sample. Compressive strength measurements (Fig. 6a) performed on both the pure hBN ceramics and HSHBN-30 show an increase in strength with increasing sintering temperature. The compressive strength of the

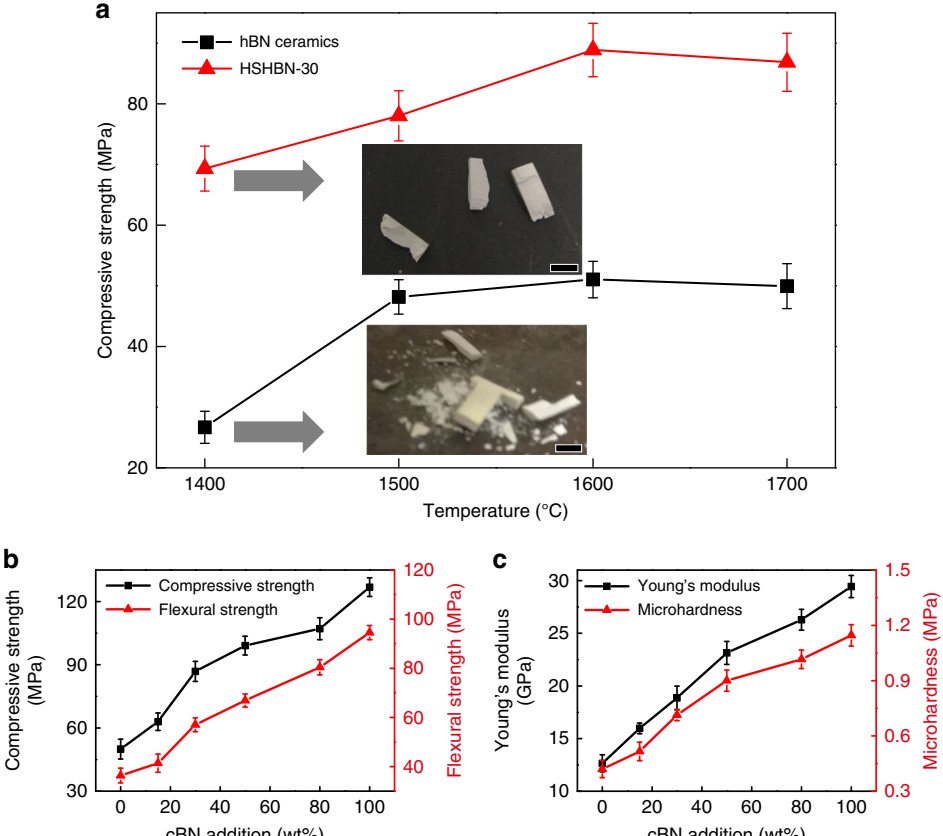

**Fig. 6** Mechanical properties of HSHBN samples. Plot showing the dependence of compressive strength on sintering temperature for both the pure hBN and HSHBN-30 ceramics, and photographs show the 1400 °C-sintered samples after testing, and all scale bars represent 6 mm (**a**); compressive and flexural strength (**b**), Young's modulus and microhardness measurements (**c**) of the 1700 °C-sintered pure hBN ceramics and HSHBNs. All the error bars represent standard deviations, and $n = 5$

1400 °C-sintered hBN ceramics is very low, and compressive strength testing of the pure hBN sample crushed the test specimen into small fragments. In contrast, the HSHBN-30 sample sintered at the same temperature only fractured into several pieces. While highly qualitative, this observation provides evidence that although the pure hBN has formed a relatively dense compact, this does not translate to a significant degree of sintering. Additionally, although the HSHBN sample with 30 wt% cBN had a lower relative density, mechanical testing indicate that it has undergone a more significant degree of sintering.

The nanoindentation test results are shown in Fig. 6c. A dramatic improvement in mechanical properties is apparent with the introduction of cBN. The sample composed of 100% cBN (HSHBN-100) exhibited the highest Young's modulus and microhardness of all measured samples, of 29.44 and 1.15 GPa, respectively, while the 100% hBN ceramics exhibited values of 15.97 and 0.52 GPa, respectively. These results demonstrate that the prepared HSHBNs are much stiffer and harder than the hBN ceramics consolidated by conventional means. The multiple curved layers forming the BN onions must possess extremely high strength to counteract the intense tensile stresses they experience, making the BN onions very strong, like carbon onions[12]. This behaviour contributes to the increased Young's modulus of the HSHBN. For conventional hBN ceramics, indentation testing may cleave the (002) crystal planes, due to the weak van der Waals forces between (002) planes, while much more energy is required to initiate fracture in the (002) layers of the BN onions in the HSHBN bulks.

To determine the relative strengths of the BN onions and hBN flakes, modelling and analysis of both morphologies were

performed with ANSYS software (Fig. 7). The Young's modulus and Poisson's ratio of each layer in the two models are referenced to the values of boronitrene, which are approximately 0.85 TPa and 0.21[25,26], respectively. Uniaxial pressure of 100 MPa was applied perpendicular to force surfaces with a radius of 15 mm on the models to analyse the strain and stress distribution, and the support surfaces are the entire bottom surfaces of the both models.

As shown in Fig. 7c, d, the equivalent elastic strain of BN onion is far less than hBN flake. What's more, the stress on the hBN flake is mainly distributed inside the body and ranges from 128.25–192.37 MPa (Fig. 7e). This onion morphology is distributed in the body evenly, and only experiences stresses of 30.71–81.07 MPa (Fig. 7f) inside the onion, approximately one-third of that in a flake. Here, we use a bigger size model to simplify the calculation, but the obtained the ratio and distribution of stresses remains and are reliable compared to the actual BN onion and hBN flake because the onion and flake are sufficiently large, and continuum mechanics apply. Therefore, hBN flakes are more susceptible than BN onions (although deformed) to damage because of the huge stress.

Due to the morphology of the BN onion being composed of multiple curved BN layers, the compressive strength in the direction parallel to the plane and flexural strength in the perpendicular direction increase significantly with the cBN addition, as shown in Fig. 6b. The weak interlayer van der Waals forces facilitate easy cleavage between the hBN layers, leading to the low strength of hBN ceramics consolidated by traditional mean. In our research, the compressive and flexural strengths of

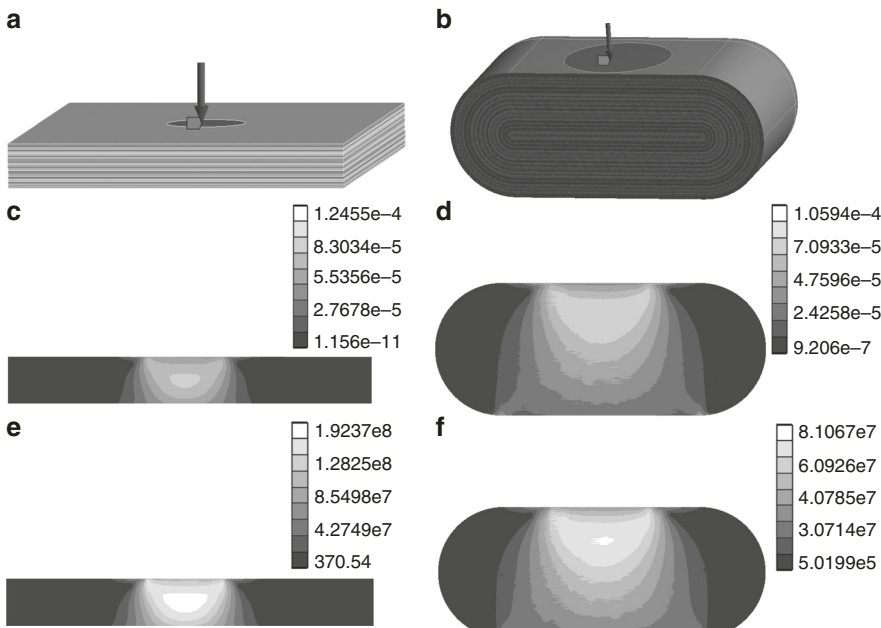

**Fig. 7** Models showing the strain and stress distributions on the hBN flake and BN onion. Models of hBN flake (**a**) and BN onion (**b**); equivalent elastic strain of hBN flake (**c**) and BN onion (**d**); equivalent stress of hBN flake (**e**) and BN onion (**f**)

pure hBN ceramics are 49.96 MPa and 36.34 MPa, respectively, confirming the previously reported results (30–50 MPa for flexural strength)[10,27]. The mechanical properties of HSHBN-30 are nearly double those of the pure hBN sample. The flexural strength of HSHBN-100 is 94.51 MPa, approximately 3 times that of pure hBN ceramics. The onion morphology ensures an intrinsically higher strength relative to layered hBN because it isn't prone to the easy cleavage that occurs in the latter material. Furthermore, some of the outer layers of the BN onions are observed to be entwined with the layers of the hBN flakes, forming an interlocking structure where the onions serve as a sort of binder.

Griffith's theory of fracture for brittle materials states that the relationship of the fracture strength ($\sigma_c$) with the elastic modulus ($E$), fracture surface energy ($\gamma$), and crack size ($c$) is described by the relation[28]:

$$\sigma_c = \sqrt{\frac{2E\gamma}{\pi c}} \qquad (1)$$

In this work, as grain growth is considered to be minimal, the enhanced bending strength of the HSHBN is primarily attributed to the effects of increased elastic modulus (Fig. 6c). Obviously, the fracture surface energy, which refers to the energy consumed to form a new surface per unit area of the material, is larger in the case of HSHBN than in pure hBN ceramics, because the fracture surface energy of curved planes of the onions are larger than the flat planes of the hBN flakes. Additionally, if the size of the BN particles is viewed as an intrinsic flaw size equal to the crack length c, which is approximately 500 nm for the BN onions and ~1–2 μm for the hBN flakes in the HSHBN bulks, a higher strength would be expected with an increasing volume fraction of onions, as the average flaw size would be significantly decreased. For the less dense HSHBNs with a high content of BN onions, as shown in Fig.6, the decrease in density does not deteriorate the mechanical properties. This demonstrates the ultra-strong innate character of the BN onions.

Figure 8 shows FESEM micrographs of the fracture surfaces of different HSHBN bulks after being fractured in bending. Compared to the pure hBN ceramics (Fig. 8a), the fracture surface of HSHBN reveals regions of both the layered structure characteristic of hBN, and the squashed morphology of the BN onions (Fig. 8e, f), with the onions being distributed between hBN flakes. It can be clearly observed that as the starting cBN content increases, more BN onions are present in the end ceramics and the preferential arrangement of the hBN flakes gradually decreases due to the onion particles preventing both the alignment and grain growth of the flakes during sintering.

## Discussion

In summary, we have pioneered a new route, to consolidate boron nitride that we call a self-densifying mechanism, which result in a new family of high strength bulk hBN materials at low temperatures using BN onions produced in situ by the addition of cBN powders, which serve as both a sintering aid and reinforcement phase. cBN particles transformed into BN onions during sintering at 1700 °C for 5 min with an increase in volume that served to fill the voids between larger hBN flakes, to produce dense bulk hBN materials with an ultra-high relative density of 97.6%. The as-prepared whole-BN bulks have a microstructure with BN onions distributed among hBN flakes, which bestows these materials with superior mechanical properties compared to pure hBN ceramics. This material design and processing methodology may be important in many other material systems. Preliminary results of thermal conductivity and dielectric properties of the as-prepared samples are consistent with those in previous reports[5,29,30] (Supplementary Note 5).

## Methods

**Sample preparation**. hBN powder (99.9%, 1–2 μm, Macklin Biochemical Co., Ltd., China) and two types of cBN powders (99.5%, average particle size of 200 nm and 400 nm, Funik Ultrahard Material Co., Ltd., China) were used as raw materials. Prior to use, the powders were treated for 10 h in a 5 wt% HCl solution to remove $B_2O_3$ and other impurities. The different powder compositions were mixed in the proportions shown in Table 2 in absolute ethanol in a beaker by ultrasonic agitation for 30 min, and then subsequently dried by heating under stirring for 2 h.

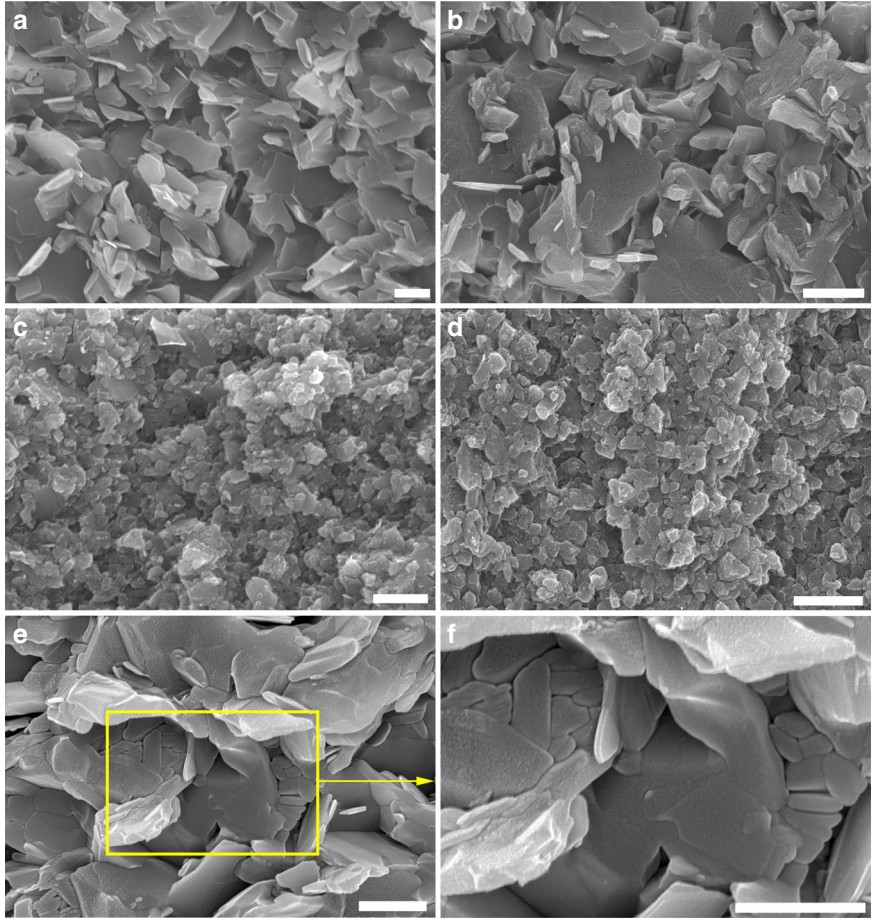

**Fig. 8** FESEM micrographs of the as-prepared samples. The fracture surfaces of pure hBN ceramics (**a**), HSHBN-30 (**b**), HSHBN-80 (**c**) and HSHBN-100 (**d**); squashed BN onions in HSHBN-30 (**e**, **f**). All scale bars represent 1 μm

The product was then placed in an oven at 70 °C for 10 h to evaporate residual solvent, and then sieved through a 120 mesh screen.

The obtained mixtures were subsequently hot pressed at 1700 °C under an applied pressure of 80 MPa on an SPS apparatus (Dr. Sinter 2040, Sumitomo Coal Co. Ltd., Japan), while HSHBN-30 and pure hBN ceramics were also consolidated at temperatures of 1400 °C, 1500 °C and 1600 °C for 5 min, at a heating rate of 150 °C min$^{-1}$.

**Characterization**. The sintering shrinkage curves were derived from the curves of the pressure head displacement vs time/temperature from 1000 °C upwards, where the thermal expansion of graphite dies and samples were taken into consideration[31].

The Archimedes' principle density measurements were employed to determine the relative densities of all samples. The phase composition of all samples was determined using XRD (Bruker D8 Advance) with Cu-Kα radiation with a step size of 0.02° 2θ over a scanning range of 15–85° 2θ. Microstructural characterization was performed using FESEM (Hitachi S-4800) and HRTEM (JEOL 200CX) at an accelerating voltage of 200 kV.

FTIR spectra were recorded on a Bruker TENSOR27 spectrometer with sample material embedded in KBr disks. Grind the samples thoroughly with an agate mortar. Transfer weighed amounts of sample powder (particle size <10 μm) approximate 1 mg, and KBr (GR) dry powder approximate 200 mg. The two powders were mixed evenly and ground thoroughly while baked with an infrared heat lamp. Transfer the ground mixture into a mold with 10 mm inner diameter and pressurize to 10 MPa, then remove the pressed translucent disk. The spectrometer runs the background channel without a sample, then puts the disk and runs the sample channel. The scanning range is 3000–400 cm$^{-1}$.

**Mechanical performance test**. Nanoindentation testing was carried out on a Bruker UMT TriboLab nanoindentation device equipped with a Berkovich pyramidal indenter. The three-point bending strength was measured on a universal testing machine (UTM, AGS-10KNG), and a crosshead speed of 0.1 mm min$^{-1}$.

The models of the hBN flake and BN onion were built by SolidWorks software (Supplementary Note 3). The equivalent elastic strain and equivalent stress analyses of the models were carried out using ANSYS software.

## Data availability

The data that support the findings of this study are available from the corresponding authors upon reasonable request.

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

## Acknowledgements

This work was supported by the National Key R&D Program of China under Grant No. 2017YFB0703201 and Natural Science Foundation of China (No.51432004 and 51672041). The authors would like to thank the State Key Laboratory of Marine Resource Utilization in South China Sea for the funding support (2016011).

## Author contributions

J.L. conceived the idea. H.T.Y. designed and performed the experiments and analyzed the results. H.F., Y.C., L.W. and W.J. contributed to the sample preparation and characterization. H.Y. performed the modeling by ANSYS software. J.L., L.W. and Y.W. explored the fabrication mechanisms. H.T.Y. wrote the manuscript with J.L., L.W. and Y.W. All authors contributed to the manuscript.

## Additional information

**Competing interests:** The authors declare no competing interests.

