## [Peer Review File · Nature Communications]

Reviewers' comments:

Reviewer #1 (Remarks to the Author):

hBN ceramics are indispensable materials and widely used in the industry. High sintering temperature and poor mechanical strength indeed restrict their further applications. Li et al. in this work developed a simple and novel approach to obtain pure and dense hBN ceramics at relatively lower sintering temperature with better mechanical performance. I could see the potentials for this approach to be explored to the other ceramic and/or alloy systems, as long as an irreversible phase transformation plus volumetric expansion coexists. Therefore, the manuscript should attract a general interest for a broader community. The work could be accepted by Nature Communication when the following concerns are properly addressed.

1. Considering the hBN ceramics were consolidated by a new approach with in-situ formed BN onions as the reinforcement, apart from their mechanical properties, other properties, e.g. thermal and dielectric property should also be reported.
2. As highlighted, cBN was introduced into the hBN matrix in this work. The authors argued that densification process during SPS was accelerated by the volume expansion caused by the cBN to hBN phase transformation. If it is true, we should see an enhanced shrinkage rate during SPS when the phase transformation occurred. Therefore, it is better to add the displacement curve in the manuscript to verify this point.
3. In the abstract, page 1 line 16, hBN ceramics haven't been considered for using at ultra-high temperature.
4. Page 5, line 86, "isomer" should be replaced by "allotrope".
5. It is not easy to distinguish BN onion from hBN flake the TEM images (Fig.3). Better to add some markers on these images. Why certain points are much brighter in the ED pattern shown in Fig.3c?
6. The way to prepare as-sintered HSHBN-30 for FTIR should be briefly explained.
7. Most of "ceramic" in the manuscript should be replaced by "ceramics".
8. Page 23, line 358, "discussion" should be replaced by "conclusion".

Reviewer #2 (Remarks to the Author):

The paper reports that a hBN material with 97.6% relative density was achieved by using cBN/hBN powder mixture and spark plasma sintering (SPS) technique, and taking advantage of the volume expansion during transformation of cBN particles to BN onions. It has been well known that BN is one of the most typical hard-to-sinter materials, and thus the densification of this material is one of the most critical issues in materials community. This paper, therefore, has certainly reached the standard of publication in other peer-reviewed technical journals on materials sciences.

However, I am not fully sure that the manuscript is suitable to Nature Communication, which requires truly deep impact and totally new originality in its publication. The work is based on the combination of the two effects of SPS and volume expansion of cBN. The former has been frequently used for densification of BN, and 95% relative density has been attained in Ref. 14 using this technique (though the temperature is substantially high). The latter effect is also a well-known behavior in transformation of cBN.

Reviewer #3 (Remarks to the Author):

This paper reports an impressive and important breakthrough in research of hBN ceramic with influences beyond this field, and should be considered to be published in Nat. Commun. The

reported unconventional and successful idea may enlighten some researchers in a puzzle to think outside box.

Dense hBN ceramic is needed in many critical instances. People have been longing for but at present are unable to produce dense bulk specimens by sintering hBN powders without sintering additives, because of the flake structure and general inertness. The authors have now overcome the difficulty via a new self-densifying mechanism. Namely, cubic BN particles among hBN flakes bring about a concurrent volumetric expansion with them transforming to form BN onions to fill the voids between hBN flakes.

In my viewpoint, this manuscript is well prepared and free of major errors. Before publication, several concerns need to be addressed.

1. Information. It is better to give readers in the Supplementary material some details from references cited in this submission about the transformation of cBN to hBN. People usually know the cutting tools of sintered cBN but do not know much of the conditions for transformation of cBN to hBN.
2. Boron oxide. The boron oxide content in the raw material is important. Any residual boron oxide after acid washing?
3. Phase transformation. The authors claim the transformation of cBN to hBN takes place from the cBN particle surface and develops inwards. Is there any possibility of the transformation of cBN to hBN happening from the core of cBN particles?
4. Growth. The authors say the growth of hBN flakes is not seen, with which I agree given the sintering conditions. How about the growth of BN onions produced in this work? Are they stable at high temperatures?
5. Data. It is appreciated to present some data on the thermal transport properties and dielectric properties to give the readership a full view of the as-prepared material. As this paper focuses on the preparation and mechanical properties, these data can be included in the Supplementary material.
6. Language. The manuscript should be double-checked to eliminate some inaccurate language uses.

Reviewer #4 (Remarks to the Author):

Ultra-low temperature self-densification of high strength bulk hexagonal boron nitride

This manuscript claimed that the density and mechanical properties of sintered hBN body were significantly improved by adding cBN particles to hBN source powder.

However, this conclusion cannot be verified from experimental results.

The authors claim that cBN was transformed to hBN at 1700°C according to XRD results. However, the sensitivity of XRD to detect cBN is low. A few weight % of cBN cannot be detected by XRD. In literature, for example, cBN did not transform to hBN at 1700°C by SPS with appropriate binder. The transformation of cBN significantly depend on density. If the cBN containing body was well densified, cBN may not transformed to hBN. (see eg. J. Zhang et al. *Ceramics International* 38 (2012) 351–356.)

If a few wt% of cBN remained in the body, the density should be apparently significantly high. The authors presented microstructure of hBN bodies in Fig.8 (a)-(d). They stress that Fig.8 (b), 30wt%cBN added, has 97.6% relative density. 97.6% is very dense; however, it does not look like 97.6%, looks much less. The authors claim that Fig.8(c) is 94 % relative density; however it also does not look like 94%. They claim that Fig.8 (d) is 87% relative density; however, it looks almost

same as Fig.8(c) 94%.

The density of this paper should be more carefully considered.

The authors claim that the strength and hardness improved with increasing cBN content. The density was decreased with increasing cBN content at more than 30wt%cBN. Generally mechanical properties, in particular hardness, significantly decrease with density. However, all mechanical properties increased with increasing cBN content. The strength and micro-hardness might have been affected by both density and microstructure, and the microstructure might have changed with cBN content. Macroscopic microstructure (such as pores and textures) on polished surface should have been presented.

The authors claim that the decrease from 2000°C to 1700°C for sintering temperature is ultra-low temperature. It is exaggerating.

The content of this paper may be inappropriate to nature communication, and appropriate to common ceramic journals.

Reviewer #1 (Remarks to the Author):

hBN ceramics are indispensable materials and widely used in the industry. High sintering temperature and poor mechanical strength indeed restrict their further applications. Li et al. in this work developed a simple and novel approach to obtain pure and dense hBN ceramics at relatively lower sintering temperature with better mechanical performance. I could see the potentials for this approach to be explored to

the other ceramic and/or alloy systems, as long as an irreversible phase transformation plus volumetric expansion coexists. Therefore, the manuscript should attract a general interest for a broader community. The work could be accepted by Nature Communication when the following concerns are properly addressed.

1. Considering the hBN ceramics were consolidated by a new approach with in-situ formed BN onions as the reinforcement, apart from their mechanical properties, other properties, e.g. thermal and dielectric property should also be reported.
2. As highlighted, cBN was introduced into the hBN matrix in this work. The authors argued that densification process during SPS was accelerated by the volume expansion caused by the cBN to hBN phase transformation. If it is true, we should see an enhanced shrinkage rate during SPS when the phase transformation occurred. Therefore, it is better to add the displacement curve in the manuscript to verify this point.
3. In the abstract, page 1 line 16, hBN ceramics haven't been considered for using at ultra-high temperature.
4. Page 5, line 86, "isomer" should be replaced by "allotrope".
5. It is not easy to distinguish BN onion from hBN flake the TEM images (Fig.3). Better to add some markers on these images. Why certain points are much brighter in the ED pattern shown in Fig.3c?
6. The way to prepare as-sintered HSHBN-30 for FTIR should be briefly explained.
7. Most of "ceramic" in the manuscript should be replaced by "ceramics".
8. Page 23, line 358, "discussion" should be replaced by "conclusion".

Reply: Thanks to the reviewer for the time and hard working on our submission. We have improved the manuscript in light of the comments.

1. The thermal and dielectric properties data of HSHBNs have been added in the Supplementary material (Fig. S4, S5).
2. The sintering shrinkage curves of hBN ceramics and HSHBN-30 (Fig. 5c) have been added and a short explanation is also added (in red in this version).

"The distinct feature is the significantly smaller overall shrinkage of the denser HSHBN-30 (~16%) compared to that of the less dense, pure hBN (~30%), strongly demonstrating that the volumetric expansion associated with the transformation of cBN to hBN makes a major contribution to the densification of HSHBN-30."

Great thanks to this reviewer as these data further confirm the proposed sintering mechanism.

3. We have changed the "ultra-high temperature" to "high temperature".

4. Yes, it is wrong to use "isomer" here, and we have changed "isomer" to "polymorph" (we found in literature this word is more often used than "allotrope" for BN), thank you.
5. We have marked BN onions and hBN flakes in Fig. 3. The brightness difference is due mainly to the size and number of involved grains in the selected area of the SAED, so it is normal to see the brightness difference. The brighter patterns are associated with the larger hBN layers while the less bright spots with a few much smaller hBN onions (they are both inside the selected area.)
6. Thank you for your valuable advice. The following passage has been added to the section of Method:
"Grind the samples thoroughly with an agate mortar. Transfer weighed amounts of sample approximate 1 mg, and KBr (GR) dry powder approximate 200 mg. The two powders were mixed evenly and ground thoroughly while baked with an infrared heat lamp. Transfer the ground mixture into a mold with 10 mm inner diameter and pressurize to 10 MPa, then remove the pressed translucent disk. The spectrometer runs the "background channel" without a sample, then puts the disk and runs the "sample channel". The scanning range is 3000-400 cm^{-1} ."
7. We are sorry for our negligence, and we have modified this word, thank you.
8. Thanks. In most journals, it is common to use "conclusion" as the last subheading. When we referred to the papers published in *Nat. Commun.*, we found the current writing style is acceptable.

Reviewer #2 (Remarks to the Author):

The paper reports that a hBN material with 97.6% relative density was achieved by using cBN/hBN powder mixture and spark plasma sintering (SPS) technique, and taking advantage of the volume expansion during transformation of cBN particles to BN onions. It has been well known that BN is one of the most typical hard-to-sinter materials, and thus the densification of this material is one of the most critical issues in materials community. This paper, therefore, has certainly reached the standard of publication in other peer-reviewed technical journals on materials sciences.

However, I am not fully sure that the manuscript is suitable to Nature Communication, which requires truly deep impact and totally new originality in its publication. The work is based on the combination of the two effects of SPS and volume expansion of cBN. The former has been frequently used for densification of BN, and 95% relative

density has been attained in Ref. 14 using this technique (though the temperature is substantially high). The latter effect is also a well-known behavior in transformation of cBN.

Reply: Thanks to the reviewer for the importance and novelty credited to our work.

Yes, it is true that the phase change of cBN to hBN is well known, which has, however, never found applications or significances in any fields, and has been a harmful tendency to be avoided when sintering cBN particles. The researchers in this report for the first time turn the tide and get off the beaten track to produce the nearly fully dense hBN ceramics that had been long-sought but not successfully produced by the powder technology.

It is true that hBN ceramics of a 95% density could be achieved by SPS hBN powders at 2000 degree, but denser samples are highly wanted which materials researchers think are very difficult, if not impossible, to produce from hBN powders. According to the previous researches, it is estimated that, to produce hBN ceramics with a density 97%, the needed operating temperature may be as high as 2300°C (In fact, till now, there is no report of the successful preparation of dense hBN ceramics at a high temperature of or above 2300°C).

Therefore, the reported results here are unbelievably encouraging and may bring new insights into the powder technology. This is the first time that sintering is associated with a volumetric expansion phenomenon rather than the pure shrinkage.

Further, the excellent mechanical properties of BN onions for the first time are incorporated into hBN ceramics, which has never been expected. All these achievements are based upon a new and creative strategy and a new sintering mechanism involving a volumetric expansion besides the prevailing volumetric shrinkage only process, which should attract “a general interest for a broader community”.

Reviewer #3 (Remarks to the Author):

This paper reports an impressive and important breakthrough in research of hBN ceramic with influences beyond this field, and should be considered to be published in *Nat. Commun.* The reported unconventional and successful idea may enlighten some researchers in a puzzle to think outside box.

Dense hBN ceramic is needed in many critical instances. People have been longing for but at present are unable to produce dense bulk specimens by sintering hBN

powders without sintering additives, because of the flake structure and general inertness. The authors have now overcome the difficulty via a new self-densifying mechanism. Namely, cubic BN particles among hBN flakes bring about a concurrent volumetric expansion with them transforming to form BN onions to fill the voids between hBN flakes.

In my viewpoint, this manuscript is well prepared and free of major errors. Before publication, several concerns need to be addressed.

1. Information. It is better to give readers in the Supplementary material some details from references cited in this submission about the transformation of cBN to hBN. People usually know the cutting tools of sintered cBN but do not know much of the conditions for transformation of cBN to hBN.

2. Boron oxide. The boron oxide content in the raw material is important. Any residual boron oxide after acid washing?

3. Phase transformation. The authors claim the transformation of cBN to hBN takes place from the cBN particle surface and develops inwards. Is there any possibility of the transformation of cBN to hBN happening from the core of cBN particles?

4. Growth. The authors say the growth of hBN flakes is not seen, with which I agree given the sintering conditions. How about the growth of BN onions produced in this work? Are they stable at high temperatures?

5. Data. It is appreciated to present some data on the thermal transport properties and dielectric properties to give the readership a full view of the as-prepared material. As this paper focuses on the preparation and mechanical properties, these data can be included in the Supplementary material.

6. Language. The manuscript should be double-checked to eliminate some inaccurate language uses.

Reply: Thanks to the reviewer for the time and hard working on our submission. We have improved the manuscript in light of the comments.

1. We have added more details of the transformation in the Supplementary materials.
2. Thank you for your valuable advice. After acid washing, no characteristic FTIR signs of B_2O_3 were detected, indicating the near zero content of B_2O_3 in the washed hBN powder.
3. Many previous reports (e.g. H. Sachdev et al. *Diam. Relat. Mater.* 6, 286–292, 1997) demonstrated the transformation takes place from the cBN particle surface. Theoretically, the phase transformation is unlikely to start from the core, as the surrounding lattices would suppress the transformation associated volumetric expansion and thus the transformation.

4. According to our observation under FESEM, the size of BN onions is generally 300-500 nm, and no growth of BN onions was found. According to previous research (e.g. J. Xue et al. *Scr. Mater.* 65, 966–969, 2011) and our work, the BN onions remain stable at 1900 °C or above.
5. The thermal and dielectric data of HSHBNs have been added in the Supplementary material (Fig.S4, S5).
6. We are sorry for our negligence and we have carefully checked the manuscript to ensure its clarity, thank you.

Reviewer #4 (Remarks to the Author):

Ultra-low temperature self-densification of high strength bulk hexagonal boron nitride

Q1. This manuscript claimed that the density and mechanical properties of sintered hBN body were significantly improved by adding cBN particles to hBN source powder. However, this conclusion cannot be verified from experimental results. The authors claim that cBN was transformed to hBN at 1700°C according to XRD results. However, the sensitivity of XRD to detect cBN is low. A few weight % of cBN cannot be detected by XRD. In literature, for example, cBN did not transform to hBN at 1700°C by SPS with appropriate binder. The transformation of cBN significantly depend on density. If the cBN containing body was well densified, cBN may not transformed to hBN. (see eg. J. Zhang et al. *Ceramics International* 38 (2012) 351–356.)

Reply: Many thanks to the reviewer for the time and hard working on our submission. In fact, both the reviewer and the authors are correct. (1) As the reviewer said, cBN particles can survive a short time at a high temperature, which makes the cBN cutting tools possible. However, here the most important factor affecting the phase transformation, particle/grain size, should not be left out. Generally, a smaller particle size is associated with a reduced transformation temperature and a faster change rate (H. Sachdev et al. *Diam. Relat. Mater.* 6, 286–292,1997). Previous research (please refer to the Supplementary materials) indicated that the transformation of fine-grained cBN can take place below 1400 °C. In the previous report cited by the reviewer (J. Zhang et al. *Ceram. Int.* 38, 351–356, 2012), cBN particles had a size of 2.8 μm, while in our work 200-400 nm. A cBN particle size 2.8 μm in Zhang et al.’s work

cited by the reviewer resulted in cBN remnant, while the size 200-400 nm in our work led to the complete transformation. The much small size of cBN used in our work ensures the complete transformation of cBN to hBN. (2) As for the influence of density on the transformation, this is actually related to the pressure. A pressure to effectively suppress the transformation needs to be tens of GPa, which is several hundreds times bigger than that in our research. Further, the compact in our work is far from dense when the transformation happens to densify the compact, and densification continues after this transformation. Namely, the density of 97.6% is the final density, not the density when the transformation proceeds. (3) The presence of phases like SiO₂, SiAlON increased the phase transformation temperature of cBN to hBN, (J. Zhang et al. *Ceram. Int.* 38, 351–356, 2012; M. Hotta et al. *J. Am. Ceram. Soc.* 92, 1684–1690, 2009), which is completely different from the whole-BN materials we prepared.

More importantly, in addition to the XRD data confirming our results, there are more evidences demonstrating the completion of this transformation in our research.

- a. TEM Our TEM data clearly show the lattice structure of onion like hBN derived from cBN particles without any cBN cores left.
- b. FTIR As reported in our submission, the FTIR data clearly tell no sign of cBN cores in the samples. The onion-like hBN particles have a radius less than 250 nm, enabling the infrared ray to transmit through the fine BN powders mixed with KBr, but no characteristic peaks of cBN were detected.

All in all, the suspicious cBN cores are actually absent in our work.

Q2. If a few wt% of cBN remained in the body, the density should be apparently significantly high. The authors presented microstructure of hBN bodies in Fig.8 (a)-(d). They stress that Fig.8 (b), 30wt%cBN added, has 97.6% relative density. 97.6% is very dense; however, it does not look like 97.6%, looks much less. The authors claim that Fig.8(c) is 94 % relative density; however, it also does not look like 94%. They claim that Fig.8 (d) is 87% relative density; however, it looks almost same as Fig.8(c) 94%. The density of this paper should be more carefully considered.

Reply: As discussed above, since the transformation is completed, no contribution to density is made by residual cBN. In this work, the pure hBN has a density of 93.6%. To say the least, if 3 wt% of cBN remained but not detected by XRD in the sample; the density would be increased 1.0% to 94.6%, not the 97.6%. As a fact, as shown in Fig.4, with temperature increase, for the same composition, the density increases

continuously while the supposed cBN remnant (if really exists) is decreased, clearly ruling out the cBN remnant viewpoint.

The reviewer is correct that the density difference is not significantly reflected in the fracture micrographs in Fig. 8. In this version, we have added some FESEM micrographs on polished surfaces in the Supplementary materials (Fig. S3), as suggested by the reviewer (not in the text where there are already FESEM micrographs on fracture surfaces of the samples.) It is true that the porosity on the surface is associated with the density, but it is not reliable to determine the density by measuring the porosity on the surface, as some smaller pores may be unseen on the surfaces. Thanks!

Q3. The authors claim that the strength and hardness improved with increasing cBN content. The density was decreased with increasing cBN content at more than 30 wt% cBN. Generally mechanical properties, in particular hardness, significantly decrease with density. However, all mechanical properties increased with increasing cBN content. The strength and micro-hardness might have affected by both density and microstructure, and the microstructure might have changed with cBN content. Macroscopic microstructure (such as pores and textures) on polished surface should have been presented.

Reply: In this version, we have added in some FESEM micrographs on polished surfaces as suggested by the reviewer (Fig. S3). The reviewer is right that the strength and micro-hardness might have been affected by both density and microstructure. As we said in the text, there are two components with their contents varying in the as-prepared hBN ceramics, say, soft hBN layers and strong squashed BN onions with the later dominating the mechanical properties. Therefore, with the increase of strong BN squashed onions and decrease of soft hBN layers, hardness and strength of the as-prepared materials increase continuously despite of a slight decrease in density. The increased contribution to strength, from the strong BN onions, more than makes up for the performance loss due to the density decrease. As said in the text, the material comprising of “pure” BN onions has the lowest density yet the greatest strength due to the robustness of squashed BN onions.

Q4. The authors claim that the decrease from 2000°C to 1700°C for sintering temperature is ultra-low temperature. It is exaggerating.

Reply: It might be inappropriate to say that the sintering temperature is decreased

from 2000°C to 1700°C because we could not produce hBN ceramics with a density 97.6% at 2000°C. According to the previous researches, it is estimated that, to produce hBN ceramics with a density 97%, the needed operating temperature may be as high as 2300°C or above. So, it might be appropriate to say that the operating temperature is decreased from 2300°C to 1700°C. But, as the reviewer suggested, we would like to modify the term of “ultra-low temperature” to “low temperature”.

Thanks!

Q5. The content of this paper is not suited to be published in *Nat. Commun.*

Reply: *Nat. Commun.* covers important achievements in all fields, including materials science. This paper reports an important breakthrough in materials science and should be taken into consideration. Yes, it is true that the phase change of cBN to hBN is well known, which has, however, never found applications or significances in any fields, and has been a harmful tendency to be avoided when sintering cBN particles. The researchers in this report for the first time turn the tide and get off the beaten track to produce the nearly fully dense hBN ceramics that had been long-sought but not successfully produced by the powder technology.

It is true that hBN ceramics of a 95% density could be achieved by SPS hBN powders at 2000 degree, but denser samples are highly wanted which materials researchers think are very difficult, if not impossible, to produce from hBN powders. According to the previous researches, it is estimated that, to produce hBN ceramics with a density 97%, the needed operating temperature may be as high as 2300°C (In fact, till now, there is no report of the successful preparation of dense hBN ceramics at a high temperature of or above 2300°C).

Therefore, the reported results here are unbelievably encouraging and may bring new insights into the powder technology. This is the first time that sintering is associated with a volumetric expansion phenomenon rather than the pure shrinkage.

Further, the excellent mechanical properties of BN onions for the first time are incorporated into hBN ceramics, which has never been expected. All these achievements are based upon a new and creative strategy and a new sintering mechanism involving a volumetric expansion besides the prevailing volumetric shrinkage only process, which should attract “a general interest for a broader community”.

After all, the comments from the reviewers are highly appreciated and our

sincere thanks go to the editor and the reviewers for the valuable time and hard working on our submission.

REVIEWERS' COMMENTS:

Reviewer #1 (Remarks to the Author):

All of my comments have been well addressed, I am happy if the manuscript could be accepted.

Reviewer #3 (Remarks to the Author):

The authors have addressed the reviewer's comments sufficiently. The paper was well organized and can be accepted for publication in Nat. Commun. now.

Reviewer #4 (Remarks to the Author):

The authors answered what reviewers asked.

It would be too difficult to show that there is no B₂O₃ or cBN.

It is interesting to use cBN to densify hBN by using the volume expansion from cBN to hBN. This results can be published on a scientific paper.

REVIEWERS' COMMENTS:

Reviewer #1 (Remarks to the Author):

All of my comments have been well addressed, I am happy if the manuscript could be accepted.

Response: Thank you very much for the time and hard working on our submission.

Reviewer #3 (Remarks to the Author):

The authors have addressed the reviewer's comments sufficiently. The paper was well organized and can be accepted for publication in Nat. Commun. now.

Response: Thank you very much indeed for your valuable time and comments on our submission.

Reviewer #4 (Remarks to the Author):

The authors answered what reviewers asked.

It would be too difficult to show that there is no B_2O_3 or cBN.

It is interesting to use cBN to densify hBN by using the volume expansion from cBN to hBN.

This result can be published on a scientific paper.

Response: Many thanks for the time and hard working on our submission. Yes, it is difficult to detect a tiny amount of cBN or B_2O_3 using XRD only. So in our work, besides the XRD tests that show the absence of cBN in the samples, the FTIR and

TEM tests were also carried out, which explicitly demonstrate that there is no cBN remains in the sintered samples.

Acid washing is an effective method to remove B_2O_3 in the raw material. As expected, after acid washing, no characteristic FTIR signs of B_2O_3 were detected, confirming the zero (or nearly zero if any) content of B_2O_3 in the samples.